# Sequence, structure prediction, and epitope analysis of the polymorphic membrane protein family in *Chlamydia trachomatis*

Patrick W. Cervantes[1][¤], Brent W. Segelke[1], Edmond Y. Lau[1], Beverly V. Robinson[1], Abisola Abisoye-Ogunniyan[1], Sukumar Pal[2], Luis M. de la Maza[2], Matthew A. Coleman[1], Patrik D'haeseleer[1] *

1 Biosciences and Biotechnology Division, Lawrence Livermore National Laboratory, Livermore, California, United States of America, 2 Department of Pathology and Laboratory Medicine, University of California Irvine, Irvine, California, United States of America

¤ Current address: Adverum Biotechnologies, Redwood City, California, United States of America
* dhaeseleer2@llnl.gov

## Abstract

The polymorphic membrane proteins (Pmps) are a family of autotransporters that play an important role in infection, adhesion and immunity in *Chlamydia trachomatis*. Here we show that the characteristic GGA(I,L,V) and FxxN tetrapeptide repeats fit into a larger repeat sequence, which correspond to the coils of a large beta-helical domain in high quality structure predictions. Analysis of the protein using structure prediction algorithms provided novel insight to the chlamydial Pmp family of proteins. While the tetrapeptide motifs themselves are predicted to play a structural role in folding and close stacking of the beta-helical backbone of the passenger domain, we found many of the interesting features of Pmps are localized to the side loops jutting out from the beta helix including protease cleavage, host cell adhesion, and B-cell epitopes; while T-cell epitopes are predominantly found in the beta-helix itself. This analysis more accurately defines the Pmp family of *Chlamydia* and may better inform rational vaccine design and functional studies.

## 1. Introduction

*Chlamydia trachomatis* is the most common sexually transmitted bacterial infection in humans with an estimated prevalence of 1–2% in the United States and 4.2% globally [1–3]. However, this estimated prevalence is likely severely underestimated as nearly 50% of male and 80% of female *Chlamydia* infections are asymptomatic and consequently go unreported [1]. *C. trachomatis* infection poses a significant health risk to humans as untreated infections can lead to blindness, pelvic inflammatory disease, ectopic pregnancy, and infertility [1, 4]. *C. trachomatis* serovars are classified into three distinct groups, or biovars, based on tissues sites of isolation and disease outcomes. These include the etiological agents of ocular trachoma (serovars A, B, Ba, C), urogenital infection (serovars D, Da, E, F, G, Ga, H, I, Ia, J, K), and lymphogranuloma venereum (LVG) disease (serovars L1, L2, L2a, L3) [5–8]. Of the different *C.*

**Funding:** This work was performed, in part, under the auspices of the U.S. Department of Energy by Lawrence Livermore National Laboratory under Contract DE-AC52-07NA27344 (PWC, BWS, EYL, BVR, AA-O, MAC, PD). Support was provided by NIH/NIAID grant U19AI144184 (PWC, BWS, BVR, AA-O, SP, LMM, MAC, PD). The funders had no role in study design, data collection and analysis, decision to publish, or preparation of the manuscript.

**Competing interests:** The authors have declared that no competing interests exist.

*trachomatis* serovars, D, E and F are known as the most prevalent in humans [5, 9, 10]. Historically, the different serovars were distinguished from one another by immunotyping using antibodies to the Major Outer Membrane Protein (MOMP) [5, 11–13]. Direct lifetime medical cost of *C. trachomatis* infections were estimated to be around $691 million in the US alone in 2018 [14]. Treatment options rely on antibiotics for known exposures or symptomatic infection, but the high prevalence of asymptomatic infection necessitates the development of protective vaccines, reviewed by de la Maza et al. [1]. In short, vaccines that target chlamydial surface proteins show promise to elicit protection, decrease transmission, and prevent adverse health outcomes.

The polymorphic membrane protein (Pmp) family is a group of surface-exposed proteins in *Chlamydia* that have been highlighted as viable vaccine candidates [15–20]. Several laboratories have shown Pmps are immunogenic in naturally infected humans, as well as in non-human primates and mice infected with *C. trachomatis* or *C. muridarum*, respectively [7, 15, 21–24]. These studies showed protein subunit vaccinations with Pmp N-terminal domains were more immunogenic and suggest cross-serovar and cross-species protection against *Chlamydia*. To further support the potential use and development of a Pmp vaccine, the N-terminal domain was shown to contain novel T-cell epitopes that neutralize a vaginal *Chlamydia* challenge [25–27].

Pmps are predicted to be autotransporters, part of the Type Va secretion system in gram-negative bacteria [28, 29], characterized by 3 functional domains: (i) a secretory sequence that facilitates transport across the plasma membrane, (ii) an outer membrane β-barrel transmembrane region, and (iii) an extracellular passenger domain that carries out a number of biological roles, from enzymatic processes to pathogenesis [30]. The Pmp family is further defined by FxxN and GGA(I,L,V) tetrapeptide repeats that are concentrated in the extracellular passenger domain (see Fig 1). These motifs occur 13.6 and 6.5 times on average in *C. trachomatis* Pmps, respectively, and are thought to play a role in cellular adhesion and infectivity [15, 28, 31–35]. However, it remains unclear whether the tetrapeptide motifs directly mediate cellular adhesion or provide passenger domain structural support. Another characteristic feature of the Pmp family is the conserved PMP_M "Middle domain" (Pfam PF07548) at the C-terminal end of the passenger domain [36], the function of which is as yet unknown.

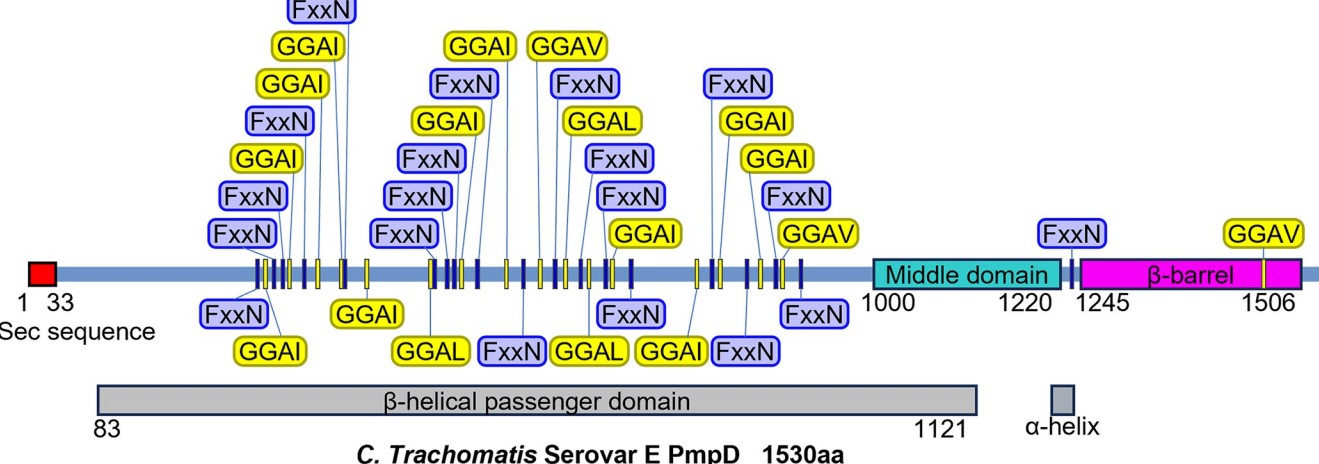

**Fig 1. *C. trachomatis* serovar E (Bour) PmpD protein linear map.** The total PmpD protein length is 1530 amino acids. The protein map shows functional domains which include a secretory sequence, the extracellular β-helix passenger domain (grey), the Middle domain (aquamarine), and the transmembrane β-barrel autotransporter (purple). The FxxN and GGA(I,L,V) tetrapeptide motifs are shown in blue and yellow, respectively. The motif repeats are concentrated in the extracellular passenger domain but can also be found, less frequently, in the C-terminal domain.

The number of *pmp* genes vary among *Chlamydia* species. *C. trachomatis* has nine *pmp* genes in three chromosomal loci, comprising more than 3% of the ~1Mbp genome [37], all of which are known to be transcribed and translocated to the outer membrane [38–40]. The nine different Pmp proteins (PmpA-I) share less than 30% amino acid identity [28] with the exception of PmpB versus PmpC (58%), whether measured across the passenger domain or the beta barrel. However individual Pmps are highly conserved across serovars, with Pmps A and D being the most conserved (~98%) and Pmp F the least conserved (~85%) [41, 42]. Interestingly, phylogenetic analysis and sequence diversity of Pmp H and F groups *C. trachomatis* serovars into 3 clades that correspond to ocular, genital, and LGV disease types [41, 43], suggesting a role for Pmps in tissue specific infection.

The passenger domain has long been predicted to form a right-handed β-helix structure [44]. A recent paper by Debrine et al [45] uses Alphafold to study the predicted β-helical structure of the passenger domain in greater depth and demonstrated the presence of many non-canonical terapetide motifs. However there is currently no crystal structure to validate the computational model of the passenger domain β-helix or the overall Pmp native structure. The *C. trachomatis* Pmp passenger domains have been shown to form hetero- and homomeric oligomers *in vitro* [34, 37, 46]. Additionally, proteolytic cleavage sites have been observed in many *Chlamydia* Pmps [33, 37, 39, 47–50], giving rise to the idea that Pmps may form diverse oligomeric structures that could contribute antigenic diversity and immune evasion [32, 34].

In this study we utilize protein structure prediction algorithms to visualize, the Pmp family from *C. trachomatis* Serovar E. In addition, our Pmp amino acid sequence analysis shows that the tetrapeptide motifs, GGA(I,L,V) and FxxN, fit into a larger and predictable spacing pattern, matching the predicted β-helical structure of the passenger domain. Our sequence and structural analysis more accurately defines the Pmp family of *Chlamydia* which could be used to inform rational vaccine design and functional studies.

## 2. Methods

### 2.1 Identifying Pmp repeats

Tetrapeptide motifs and longer motifs with various spacings in the Pmp protein sequences were investigated using the ScanProsite tool [51]. The resulting longer motif instances were aligned using the Multiple Alignment using Fast Fourier Transform (MAFFT) tool [52] with the G-INS-i option for global alignment, and the resulting alignment was then turned into an HMM model using HMMSEARCH [53] to search against the Uniprot database. The Pmp amino acid sequences used included all 9 Pmps from *C. trachomatis* serovar E (Bour), all 9 *C. muridarum* (NiggII) Pmps, and all 16 *C. pneumoniae* (TWAR) Pmps. The complete list with protein accession numbers can be found in S1 Table. The ScanProsite motif hits, and the HMM model can be found in S2 Table and S1 File.

### 2.2 Protein structure prediction

After removing the signal peptides, protein sequences for all nine *C. trachomatis* serovar E Pmps, PmpD of *C. trachomatis* serovar L2, and Pmp21 of *C. pneumoniae* were submitted for protein structure prediction using both the TrRosetta [54] and RoseTTAFold [55] algorithms through the Robetta protein structure prediction service hosted by the Baker lab at https://robetta.bakerlab.org/. The five models generated by Robetta for each sequence were inspected visually and a representative model chosen. For proteins longer than 1000aa (the length limit on the Robetta server at the time these jobs were submitted), the first and last 1000aa were submitted for structure prediction separately. The resulting models were then aligned using Matchmaker in UCSF Chimera [56], and representative models with the best structural

homology between the two partial models were spliced together into a single structure. Alpha-Fold2 structure predictions for all the proteins in the Swissprot database–including the Serovar D homologs of the Pmp proteins–have recently been made available through Uniprot or directly through the AlphaFold Protein Structure Database at https://alphafold.ebi.ac.uk/faq [57, 58], however these pregenerated structures typically include the signal peptide, which can impact the structure at the top of the passenger domain. We generated Alphafold2 structure predictions for the mature Pmp proteins without the signal peptides, using the Aphafold2 Colab notebook at https://colab.research.google.com/github/deepmind/alphafold/blob/main/notebooks/AlphaFold.ipynb.. Likewise, we generated ESMFold structure predictions [59] using the Colab notebook provided by Sergey Ovchinnikov at https://colab.research.google.com/github/sokrypton/ColabFold/blob/main/ESMFold.ipynb).

## 2.3 Prediction of membrane binding domains

We used the DREAMM public web server at https://dreamm.ni4os.eu/ to predict the membrane-penetrating amino acids in Pmp proteins. DREAMM is a recently developed ensemble machine learning algorithm for predicting protein–membrane interfaces of peripheral membrane proteins [60].

## 2.4 Molecular dynamics

All molecular dynamics simulations utilized the MARTINI coarse-grain force field (version 2, the proteins used the version 2.2 parameters) [61, 62]. The system was prepared using the CHARMM-GUI [63]. Each protein was embedded into a POPC bilayer, solvated with the MARTINI polarizable water model [64], and the appropriate number of Na + /Cl—was added to neutralize the system and obtain a concentration of 150 mM. The program GROMACS (version 2021.1) was used for the MD simulations [65]. The system was energy minimized and 1 microsecond of production dynamics was performed. Simulations were performed in the NPT ensemble with weak temperature and pressure coupling using a velocity-rescaling thermostat at 303 K (coupling constant 1.0 ps) [66] and the Parrinello-Rahman barostat (coupling constant 12 ps) [67]. A timestep of 20 fs was used. Electrostatics were calculated with a reaction field (dielectric constant of 2.5) and cutoff at 1.1 nm. The van der Waals interactions were calculated with a cutoff at 1.1 nm.

## 2.5 Epitope prediction

We used analysis tools provided through the Immune Epitope Database (IEDB) web server [68] for prediction of B- and T-cell epitopes. The B-cell epitope predictions were generated with the protein sequence based tool BepiPred-2.0 [69], as well as the structure based tool Discotope [70], the latter using our RoseTTAFold-predicted structures of the Pmps as input. T-cell epitopes predictions were generated using the recommended settings for the T-cell epitope prediction tools at IEDB, which currently default to NetMHCPan 4.1 EL for MHC-I, and Consensus 2.22 for MHC-II. For MHC-II prediction, we focus on a set of 13 HLA alleles that incorporates some of the most frequent DRB1/3/4/5 alleles, and supertype alleles that cover well-over 95% of most HLA types present in human population groups [71–73]: DRB1*01:01, DRB1*03:01, DRB1*04:01, DRB1*07:01, DRB1*08:01, DRB1*09:01, DRB1*11:01, DRB1*13:01, DRB1*15:01, DRB3*01:01, DRB3*02:02, DRB4*01:01, DRB5*01:01.

## 2.6 Pmp amino acid sequence conservation

Multiple Alignment using Fast Fourier Transform (MAFFT, v7.487) [52] was used to align *C. trachomatis* Pmp amino acid sequences from serovars A-K, Ba, and L1-L3. Alignments were

uploaded to the Multalign View tool in UCSF Chimera and the percent amino acid conservations were mapped onto the respective *C. trachomatis* Serovar E Pmp structural model generated in RoseTTAFold.

## 3. Results

### 3.1 Pmp passenger domains contain a larger repeat sequence that incorporates the GGA(I,L,V) and FxxN repeat motifs

To test the statement by Grimwood and Stephens that "the number of amino acids between the [GGA(I,L,V) and FxxN] repeat motifs does not appear to follow any characteristic spacing pattern" [28], we examined the spacing between motifs in all nine *C. trachomatis* serovar D and *C. muridarum* Pmps, and all 16 *C. pneumoniae* Pmps. We found that GGA(I,L,V) motifs are almost always followed by a FxxN motif (Fig 2A), and 75% of these pairs (169/226) are spaced only 14–18 amino acids apart (Fig 2B). The spacing following FxxN motifs is much more variable. In about 57% of the cases (122/214), an FxxN motif is followed by a GGA(I,L,V) motif with a gap of only 4-5aa, but much longer gaps or a second FxxN motif are also common. Out of the 34 Pmp proteins examined, we found 188 instances of the core pattern (G-G-A-[ILV]-x(14,18)-F-x-x-N in PROSITE notation), and 301 instances if we allowed one amino acid difference to the core pattern or allowed the gap to vary from 12–23 amino acids. Fig 2C shows the sequence logo for the HMM derived from these sequences, showing several conserved amino acid positions between the classical tetrapeptide motifs, including an asparagine at position 15 conserved in 55% of the sequences, and alternating hydrophobic amino acids at positions 4, 6, 10, 12, 19, 21, which are likely facing toward the inside of the β-helix.

Interestingly, Pfam contains a "*Chlamydia* polymorphic membrane protein (Chlamydia_Pmp) repeat" domain (PF02415) that incorporates both tetrapeptide motifs, but in the opposite order: FxxN, followed by a long and variable gap with little sequence specificity before the GGA(I,L,V) motif (Fig 2D), which omits many of the conserved positions we identified. Although this is supposedly a *Chlamydia* specific Pfam domain, less than a third of the

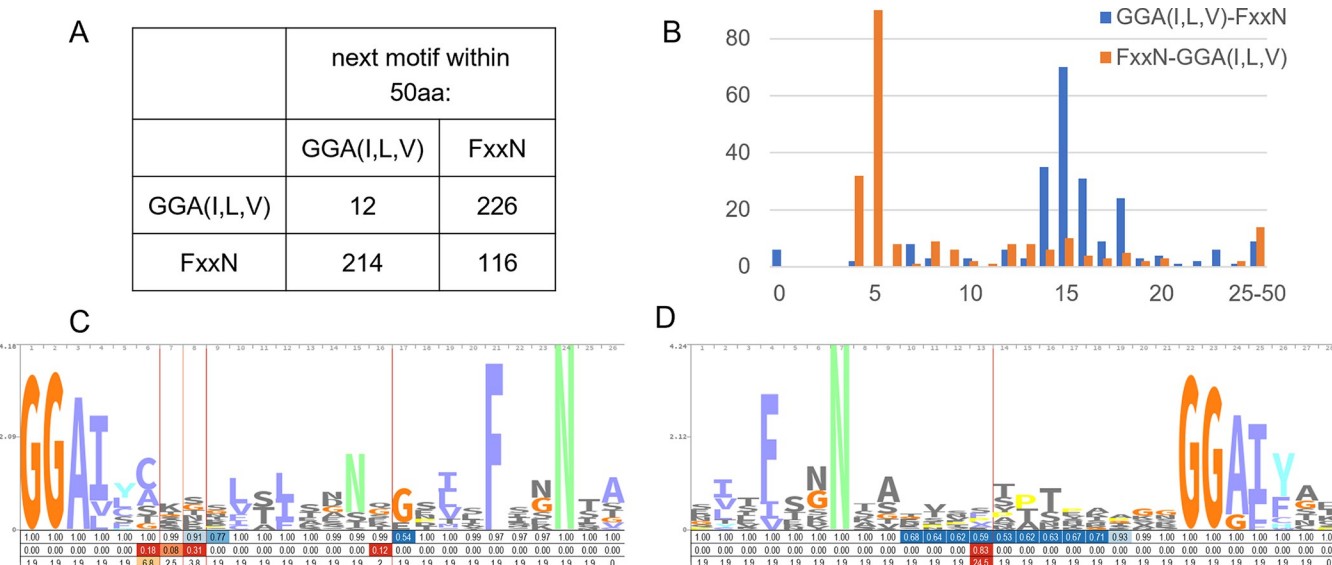

**Fig 2. The Pmp tetrapeptide repeats, GGA(I,L,V) and FxxN, fit in a larger repeat.** (A) The GGA(I,L,V) and FxxN motifs tend to alternate, with GGA(I,L,V) almost always followed by FxxN. (B) GGA(I,L,V) and FxxN repeats show a regular spacing. (C) Sequence logo for the longer repeat we identified. (D) Sequence logo for PF02415.

proteins containing this domain (443 out of 1448 Uniprot proteins based on HMMSEARCH) belong to *Chlamydia* species, including almost 200 non-bacterial proteins. In contrast, the HMM we derived here is almost exclusive to *Chlamydia*, matches a larger number of *Chlamydia* proteins (675 out of 744 Uniprot proteins), and identifies more than double the number of repeats per protein (average 6.1 repeats per protein, versus 2.9 for PF02415).

### 3.2 Structural modeling shows Pmps share many features with other autotransporters

We initially generated protein structure predictions for all *C. trachomatis* Pmps using the TrRosetta algorithm [54] available on the Baker lab's public Robetta protein structure prediction service (https://robetta.bakerlab.org/), and then upgraded to their new RoseTTAFold algorithm [55], and then DeepMind's record breaking Alphafold2 [57], and Meta's ESMFold [59] as those became publicly available. The older TrRosetta algorithm (Fig 3A) cannot resolve some regions such as the passenger domain as well as the newer algorithm, and was not used for any further analysis. The newer algorithms all converge on a similar structure (Fig 3B–3D)–a promising finding as the methods to generate models by RoseTTAFold/AlphaFold2 are very different from the large language model-based ESMFold. Although all three modelers struggle to resolve the structure of the large side loops–with the AlphaFold structures often showing large completely unstructured side loops (Fig 3C)–we chose to focus primarily on the RoseTTAFold structure predictions, because they tended to predict less unstructured side loops with slightly higher confidence scores. However without any confirming experimental data, the exact structures of the side loops is speculative. PDB files with the RoseTTAFold structure predictions for PmpA to PmpI of *C. trachomatis* serovar E can be found in S2 File.

The resulting structures show many of the stereotypical features found in other autotransporters, including a 12-stranded β-barrel at the C-terminal. The predicted β-barrel tended to incorporate multiple mortise-tenon joints [75], and a β-hairpin structure as part of the fifth

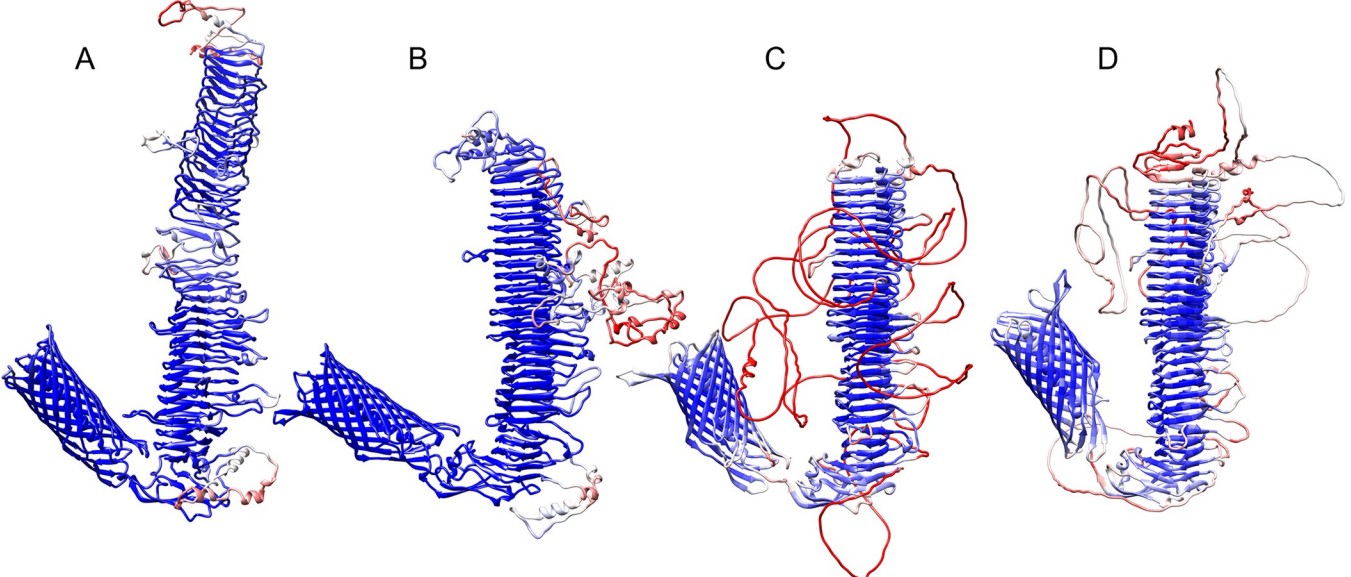

**Fig 3. Comparison of structure predictions for PmpC serovar E.** Structure predictions using TrRosetta (A), RoseTTAFold (B), AlphaFold2 (C), and ESMFold (D). Confidence scores (pLDDT for AlphaFold and ESMFold, RMSD for TrRosetta and RoseTTAFold) were mapped to the same blue-to-red scale using the empirical formula $C_\alpha RMS = 1.5 \exp(4 \times (0.7 - pLDDT))$ [55, 74].

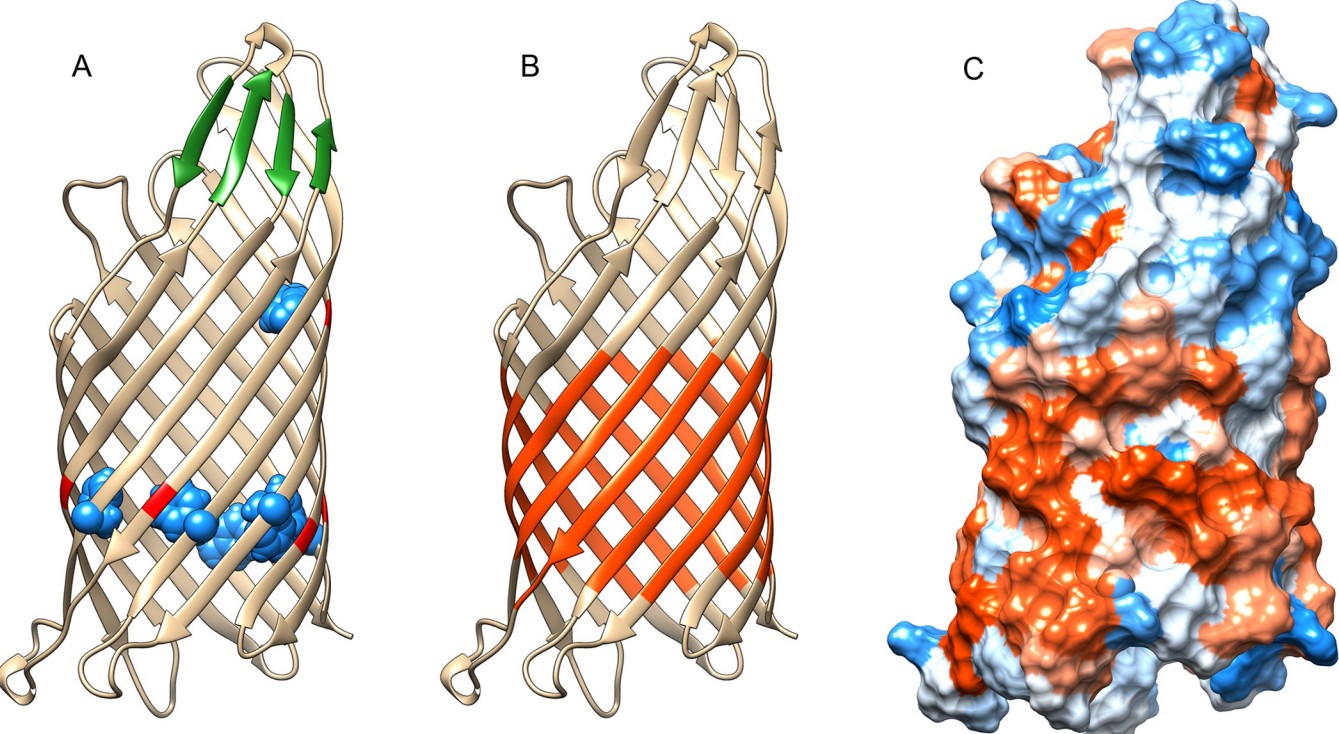

**Fig 4. The Pmps include a characteristic C-terminal transmembrane β-barrel.** (A) The β-barrel for PmpG showing six "mortise-tenon joints", in which an aromatic side chain facing into the lumen of the barrel domain locks into a void created by the presence of a glycine residue on the neighboring β-strand. Blue: aromatic side chains. Red: glycine residues. Shown in green is a 4-stranded β-sheet formed by β-hairpins in extracellular loops L4 and L5. The β-hairpin in L5 has been shown to be important for correct folding of the passenger domain in other autotransporters. Prediction of transmembrane β-strands using TMbed (B) and visualization of the hydrophobic residues (C) indicates that only the bottom 3-4nm of the β-barrel is predicted to be embedded in the Chlamydia outer membrane.

extracellular loop of the β-barrel shown to be important for correct folding of the passenger domain in other autotransporters [76] (Fig 4A). The predicted β-barrel averages approximately 16 residues per β-strand, for a total height of about 7nm, similar to known autotransporter β-barrel crystal structures such as *E. coli* EspP (PDB 3SLJ). However the width of the *Chlamydia* outer membrane bilayer is predicted to be only 4.23 nm (RB) to 4.41 nm (EB) [77], suggesting that only part of the β-barrel is embedded in the membrane. Indeed, only the lower 3–4 nm of the β-barrel is covered with hydrophobic residues (Fig 4B, 4C), suggesting that the remainder of the Pmp β-barrel protrudes extracellularly.

The conserved Pmp_M "Middle domain" is predicted to include the first couple of coils of the β-helix most proximal to the β-barrel (even though it does not contain the conserved GGA (I,L,V)/FxxN motifs or HMM sequence pattern described above). The Middle domain flattens to a β-sandwich and is capped by a β-hairpin. This structure is similar to autochaperone domains found in many autotransporters with β-helical passenger domains such as pertactin [78–80], suggesting it may play a role in proper secretion and folding.

As shown by others, the passenger domain is predicted to fold into a tightly coiled β-helix formed by parallel β-strands (Fig 5A). Based on the structure predictions shown here, Pmps and their homologs in related *Chlamydia* species may be some of the longest known parallel β-helix proteins, with at least 27 complete coils for PmpB and 22 coils for PmpD, versus 16 coils for Pertactin.

Following the nomenclature introduced by Yoder et al. 1993 and Jenkins et al. 2001 [80, 81], the three parallel β-sheets (PB) in PmpD are labeled PB1 (yellow), PB2 (blue) and PB3

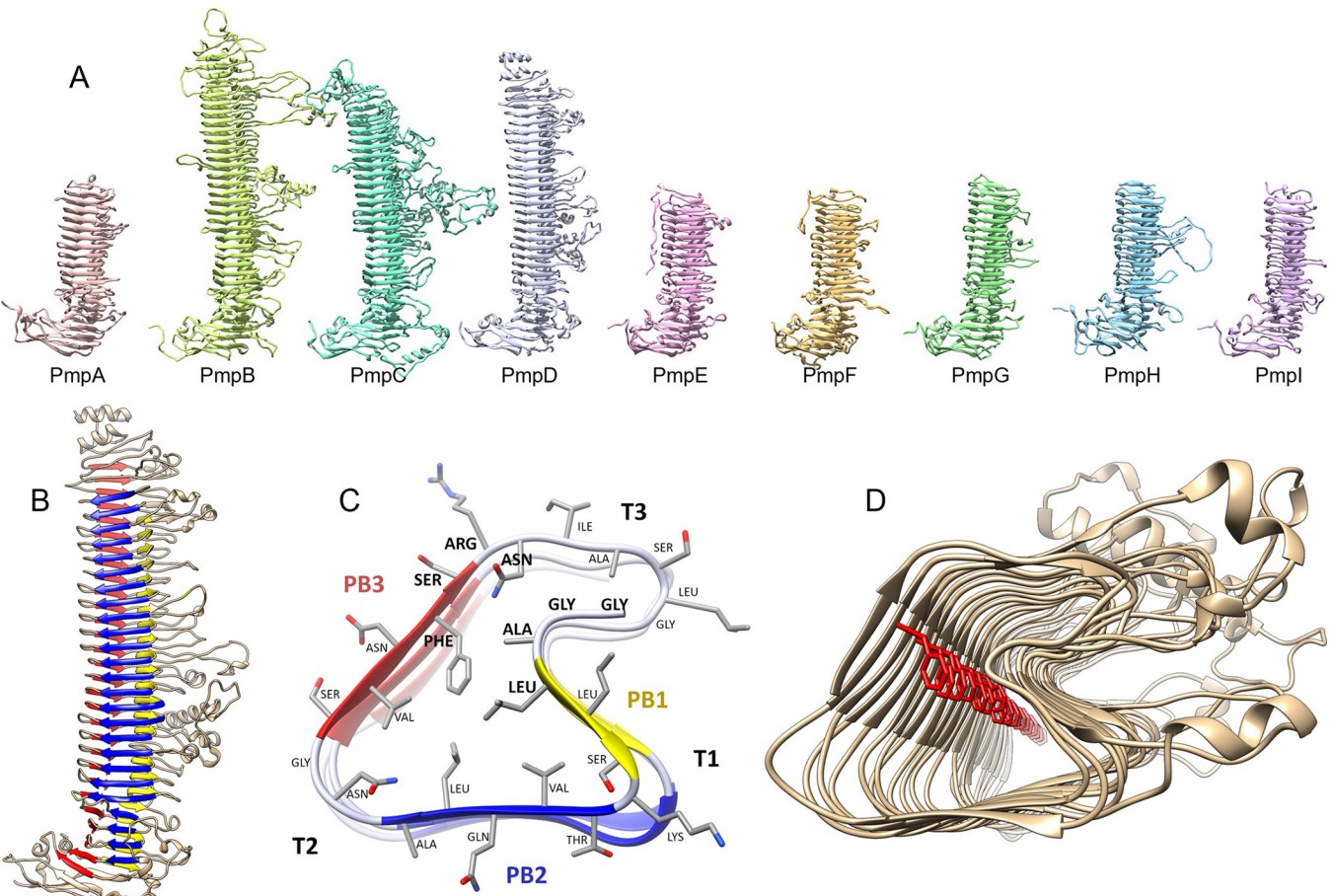

**Fig 5. Pmp passenger domains form a regular β-helical structure.** Pmp passenger domain structural models and nomenclature illustrating the parallel β-sheets (PB) and turns (T). (A) Family portrait of Pmp passenger domain models. From left to right: PmpA, B, C, D, E, F, G, H, and I. (B) PmpD passenger domain showing 22 complete β-helix coils. (C) PmpD passenger domain cross section between amino acid residues 615–641 shows how the repeat sequence forms the core of the β-helix. Note that the nomenclature used here follows the well-established numbering by Yoder et al., 1993 and Jenkins et al., 2001 [80, 81], rather than the one proposed by [45]. The parallel β-sheet PB1 is shown in yellow, PB2 in blue, PB3 in red. The GGA(I,L,V) and FxxN motifs are bolded—in this case we have GGAL at the transition from T3 to PB1, and FSRN at the transition from PB3 to T3. (D) π-stacking interactions between the conserved phenylalanines may help stabilize the core of the β-helix.

(red) and the turns (T) following the β-sheets labeled T1, T2, T3 (Fig 5C). Compared with the HMM pattern in Fig 2C, the initial conserved GGA constitutes the end of T3, followed by a short ~3 amino acid PB1 (Fig 3B). The two areas with sequence length variability between positions 6–9 and 16–17 (indicated by the vertical lines in Fig 2C) correspond to turns T1 and T2, while the conserved FxxN motif marks the transition from PB3 to T3 (Fig 3B). It is important to note that the side chain positions of the most conserved amino acids in the GGA(I,L,V) and FxxN tetrapeptide motifs point inward, within the β-helix core (Fig 5C), suggesting that the tetrapeptide motif provide a more structural role to the passenger domain β-helix. We also show that most of the variability in this pattern lies within the sometimes very long side loops inserted in T3 (also referred to as Ω-loops [45, 82]) and occasionally T1, while T2 typically only varies in length by one or two amino acids (Fig 2C).

All predicted Pmp structures showed a 12-stranded β-barrel at the C-terminal, connected to the passenger domain by an α-helix threaded through the center of the β-barrel (Fig 4). Surprisingly, most models favored an angle of 90 degrees or more between the β-helical passenger domain and the transmembrane β-barrel, rather than having the two domains in a straight

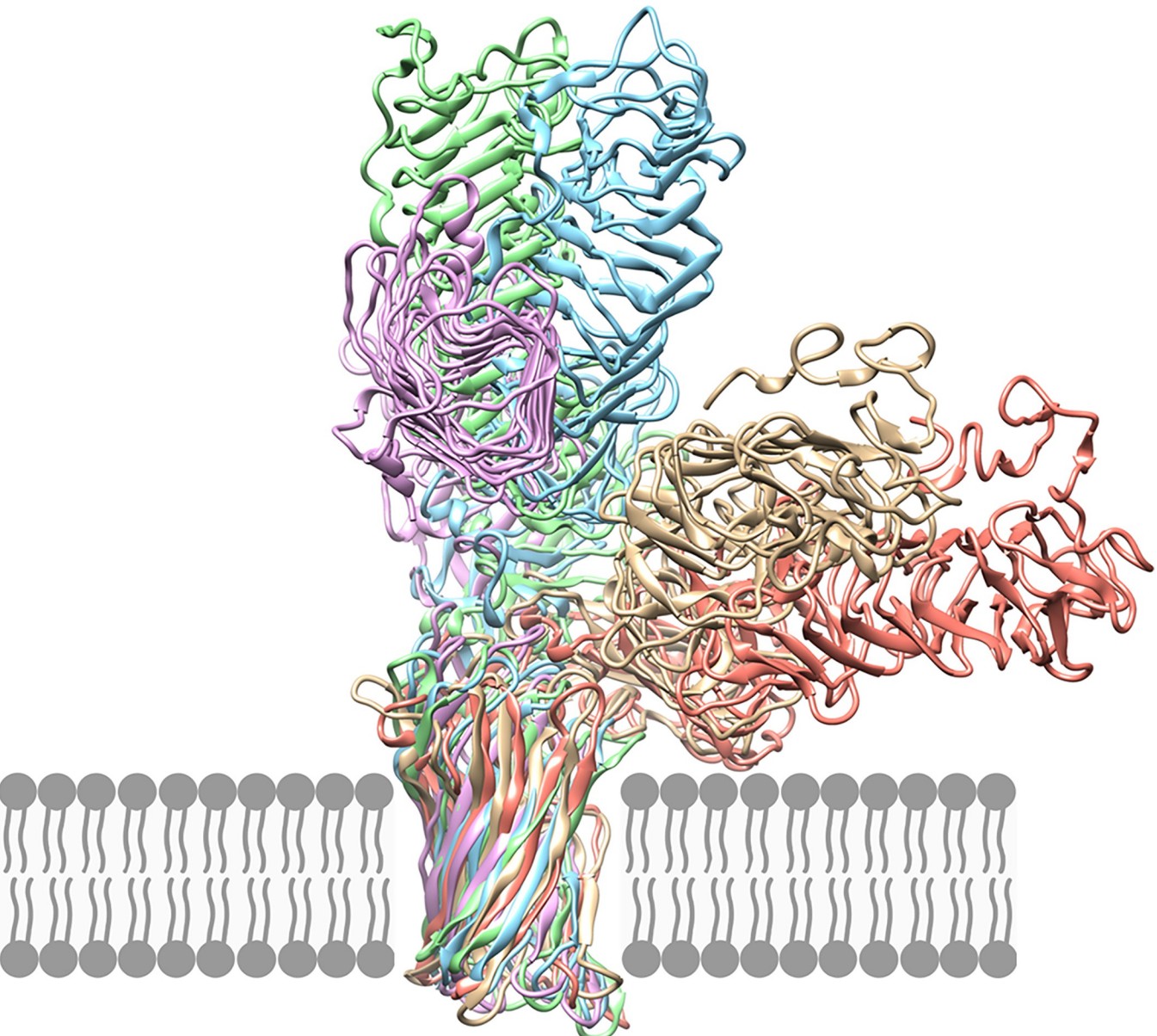

**Fig 6. The Pmp passenger domains are connected to the transmembrane β-barrel by a flexible hinge.** Coarse-grained molecular dynamics simulation of PmpE with the β-barrel domain embedded in a POPC bilayer shows that the passenger domain is connected by a flexible hinge between the β-barrel and the Middle domain.

line. Coarse-grained molecular dynamics simulations suggest a flexible hinge region between the membrane-embedded β-barrel and the base of the passenger domain, allowing the extracellular portion of the protein to take on a range of conformations, from standing straight out from the membrane to lying flat against it, as illustrated in Fig 6.

## 3.3 Protease cleavage sites are concentrated in side loops

[83] lists a total of 21 Pmp cleavage sites that have been experimentally detected in serovar L2 Pmp proteins by N-terminal sequencing [37], mass spectrometry [50], and identification of semitryptic peptides [39]. Mapping these cleavage sites to our serovar E structures, we found

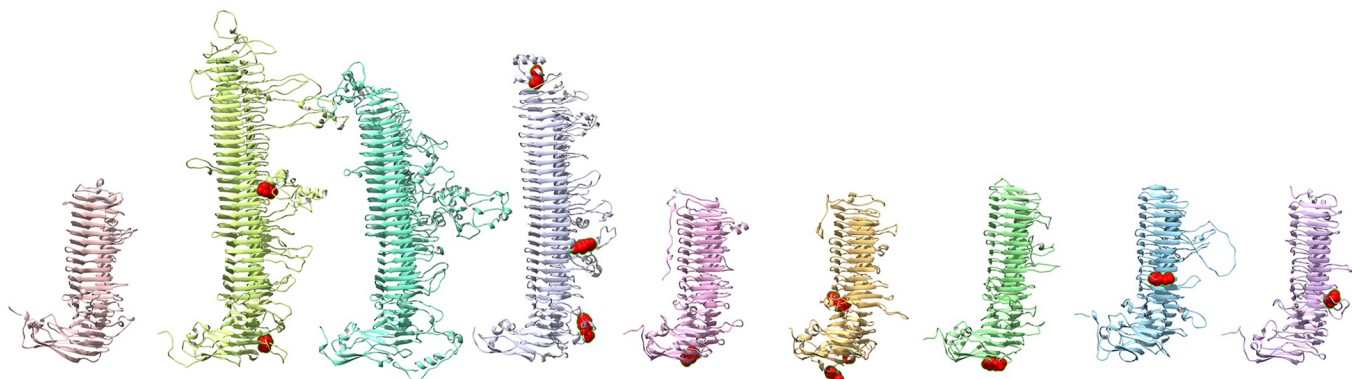

**Fig 7. Pmp cleavage sites for serovar L2 Pmp proteins mapped to the serovar E passenger domain structures.** Pmp cleavage sites from serovar L2 were mapped by BLAST to the homologous locations on the serovar E PmPs, for purposes of illustrating their location in the side loops only. Note that these cleavage sites may not be present or active in the serovar E Pmps.

that three of these sites map to signal peptide cleavage sites, two cleave the alpha helix inside the transmembrane β-barrel (PmpD and PmpG), and two cleave the β-barrel itself. Of the remaining 14 cleavage sites that fall within the passenger domains, only two are located inside the β-helix coils, while the remaining 12 map to the side loops, the N-terminal cap, or the non-β-helical parts of the Middle domain (Fig 7).

## 3.4 B-cell epitopes are concentrated in side loops, T-cell epitopes in the main β-helix of the passenger domain

The Immune Epitope Database (IEDB) [68] contains data on experimentally validated B-cell epitopes for 40 different *C. trachomatis* antigens, including 7 epitopes for PmpD and 5 for PmpC. All experimentally validated B-cell epitopes for PmpC and PmpD map to side loops of the β-helix or the Middle domain (Fig 8A). Assuming many Pmp epitopes remain to be discovered, we also utilized computational methods available through IEDB to predict additional epitopes. The first prediction method, Bepipred-2.0 [69], uses the linear sequences of proteins to predict epitopes. We input the *C. trachomatis* serovar E sequences for each member of the Pmp family (A-I). We also predicted B-cell epitopes using the protein structure based Discotope 2.0 algorithm [70] to predict B-cell epitopes for each Pmp based on surface accessibility and amino acid epitope propensity scores. Similar to the experimentally validated epitopes, epitopes predicted from either sequence or structure also localized predominantly in the passenger domain side-loops, and especially in some of the long extended loops (Fig 8B). B-cell epitope predictions on the Alphafold2 predicted structures were largely identical (data not shown). The extracellular face of the transmembrane β-barrel also scored high in the structure-based B-cell epitope prediction, even though IEDB contains no known epitopes in this area. However when taken in context of the protein in its biological setting (embedded into the membrane and shielded by the large passenger domain or other membrane components) these regions would likely not bind antibodies. B-cell epitope predictions using Bepipred-2.0 and Discotope 2.0 can be found in S3 and S4 Tables.

Experimentally validated T-cell epitopes in IEDB occur for 49 different *C. trachomatis* antigens, which include one epitope each in PmpC, D, E, G, H, and I. Two MHC-I epitopes map to different strands of β-barrel for PmpC and PmpI, while the four MHC-II epitopes in PmpD, E, G, and H map to various locations along the β-helix.

Computational prediction of MHC-II T-cell epitopes map predominantly to the β-helix of the passenger domains (Fig 8C), with some epitopes matching multiple HLA subtypes.

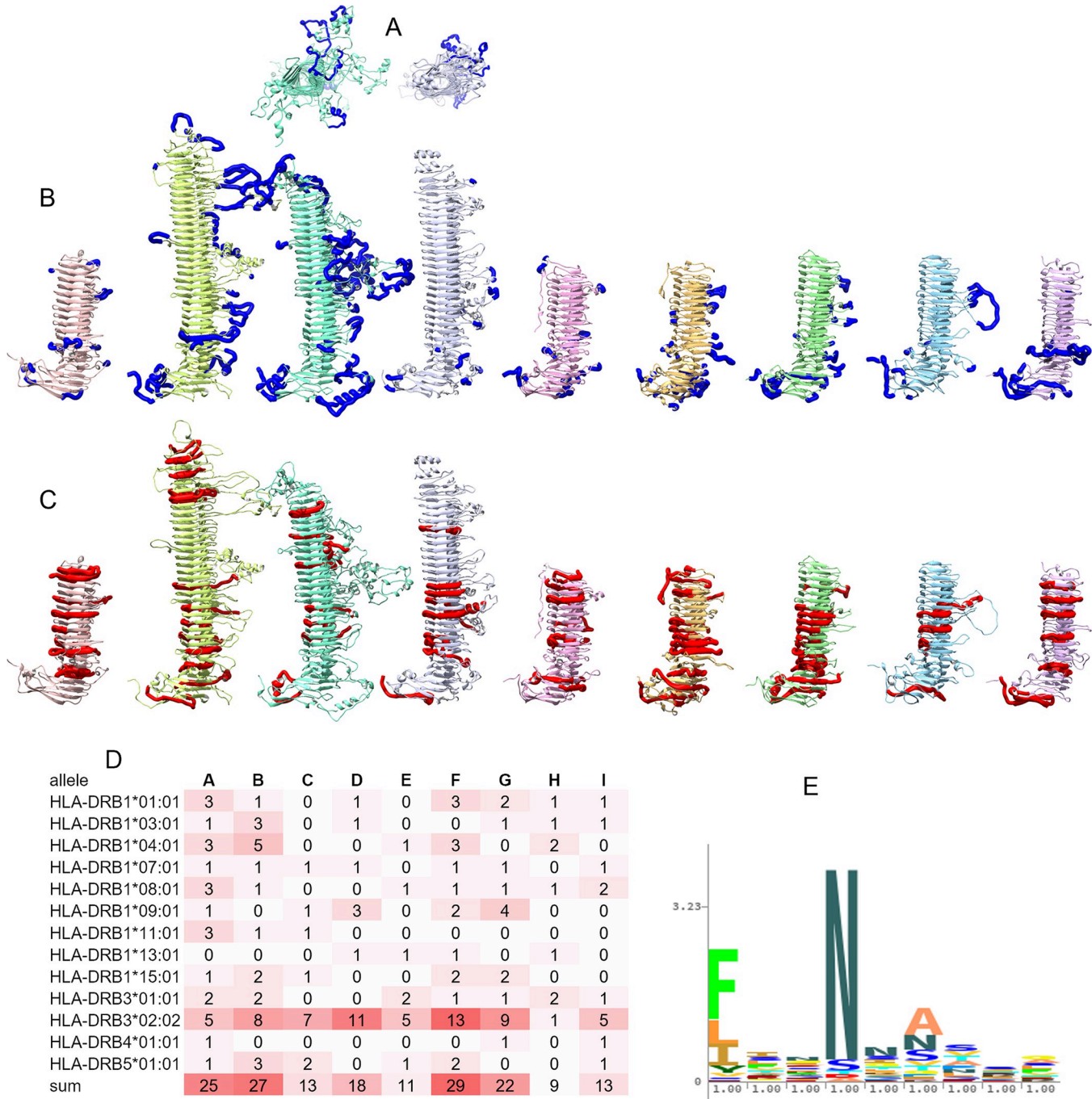

**Fig 8. B-cell and T-cell epitopes on Pmp structure.** (A) Top view of the PmpC and PmpD passenger domains, with experimentally validated B-cell epitopes. (B) Computationally predicted B-cell epitopes based on structure, using Discotope. From left to right: Pmp A, B, C, D, E, F, G, H, and I. (C) Computationally predicted MHC-II T-cell epitopes based on structure. (D) The number of MHC-II T-cell epitopes differs considerably across Pmps and HLA subtypes. Note that DRB3*02:02 is overrepresented, and some of the smaller Pmps have significantly more MHC-II epitopes than the larger ones, as indicated by the red highlighting. (E) The core binding motif for MHC-II allele HLA-DRB3*02:02 overlaps with the FxxN repeat of the β-helix.

Interestingly, epitopes for allele HLA-DRB3*02:02 seem strongly overrepresented, accounting for more than one third of all MHC-II T-cell epitopes (Fig 8D). Upon further examination, it appears that the core binding motif for this allele overlaps with the FxxN motif in the passenger domain β-helices (Fig 8E). This suggests that populations with this particular MHC-II allele

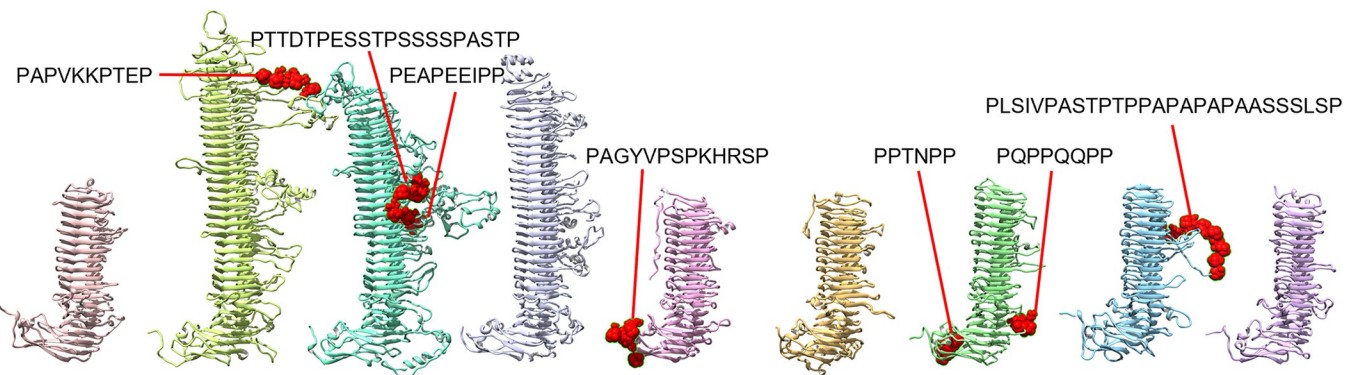

**Fig 9. Proline-rich regions in the Pmp passenger domains may be involved in adhesion.** Proline-rich regions are located primarily in the side loops and the Middle domain. Sites with at least four prolines separated by 0–4 residues are highlighted in red and labeled with the sequence.

may be able to mount a stronger T-cell immune response to *Chlamydia* infections. The number of MHC-II T-cell epitopes differs noticeable across Pmps, with the short PmpA containing 25 epitopes (20 not counting HLA-DRB3*02:02) for 12 of the 13 HLA subtypes tested, while the equally short PmpH only contains 9 (8 not counting HLA-DRB3*02:02), and the much longer PmpC contains 13 (6 not counting HLA-DRB3*02:02) (Fig 8D). This suggests that some Pmps may present better vaccine targets to stimulate a broad T-cell response. MHC-I T-cell epitopes were predicted to be fairly abundant in the Pmps and spread throughout the entire length of the protein.

MHC-II class T-cell epitope predictions using IEDB's Consensus 2.22 can be found in S5 Table.

### 3.5 Side loops are predicted to be involved in host cell adhesion

Pmp side loops include proline-rich regions (PRRs) that may be involved in binding to host membrane proteins. PRRs are often involved in eukaryotic protein-protein interactions, and can be exploited by viral and bacterial pathogens to interact with host proteins [84, 85]. A wide range of different sequence motifs have been described for PRRs, but they tend to contain at least four prolines, separated by up to four other amino acids. In the Pmps, PRRs are predominantly found on the side loops and the Middle domain (Fig 9).

Besides binding to membrane proteins, Pmps may also interact directly with the host membrane. As expected, prediction of protein-membrane interface peptides using DREAMM yields multiple hits in the transmembrane β-barrel for each Pmp. In addition, each Pmp has one or more predicted membrane-penetrating amino acids at the N terminal and in various side loops off the main β-helix, that may be involved in adhesion to host membrane (Fig 10). Membrane-penetrating amino acids predicted by DREAMM can be found in S6 Table.

Sequence variation between the *C. trachomatis* serovars, based on a multiple sequence alignment with MAFFT is also more concentrated in the passenger domain side loops (Fig 11), which may be correlated with the positive selective pressure on the B-cell epitopes by the host immune system, and with differences in tissue tropism. Multiple sequence alignment for each of the *C. trachomatis* Pmps using MAFFT can be found in S3 File.

## 4. Discussion

The Pmp family is a group of surface exposed chlamydial proteins that are naturally immunogenic in humans and may function as viable vaccine antigen candidates [15, 21–24]. In this

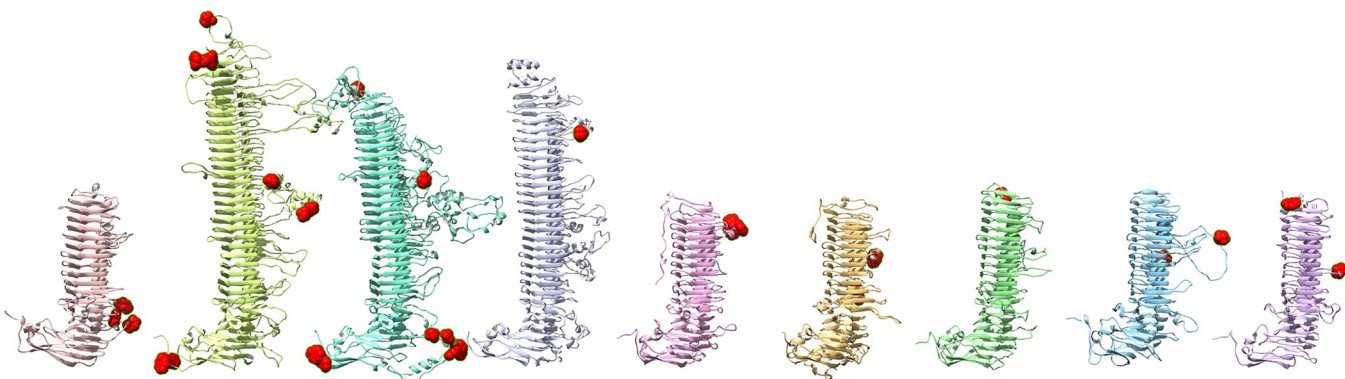

**Fig 10. Putative membrane-penetrating amino acids, predicted by DREAMM.** Predicted membrane-penetrating amino acids (highlighted in red) are located primarily in the side loops, and at the N terminal, as well as the transmembrane β-barrel (not shown).

study, we took advantage of the latest generation of protein structure prediction tools to create full-length protein structure predictions for all nine Pmp proteins (PmpA-I) from *C. trachomatis* serovar E (Bour). Although the GGA(I,L,V) and FxxN tetrapeptide motifs have been recognized as prominent features of the Pmp passenger domains for more than twenty years [28], their semi-regular spacing had not been previously appreciated. Our bioinformatic and structural analyses discovered that the tetrapeptide motifs fit into a larger repeat sequence that can be fit with an HMM model including several other partially conserved residues, and that these larger sequence repeats correspond to the individual coils of the predicted β-helical structure of the passenger domains. Since not all the β-helical coils include the canonical GGA(I,L,V) and FxxN motifs, as also pointed out by [45], it may be possible to derive a more inclusive HMM by starting from a structural alignment of the coils.

The availability of a high quality HMM also makes it trivial to identify additional Pmp-like proteins, where previous efforts had to resort to searching for the short tetrapeptide motifs [45], PSI-BLAST [86], or the use of a Pfam that misses around a third of Chlamydia's own Pmp's [87, 88]. For example, the highest scoring hit to our HMM from Fig 2C is not to a Chlamydia protein, but a pectin lyase fold/virulence factor from Akkermansia glyciniphila (accession A0A1C7PBX1, HMM E-value 3.6e-158) with 27% sequence identity to the nearest Chlamydia protein, predicted to fold into a classical autotransporter with a β-barrel and a passenger domain with a record breaking 43 β-helical coils. We also confirm the presence of

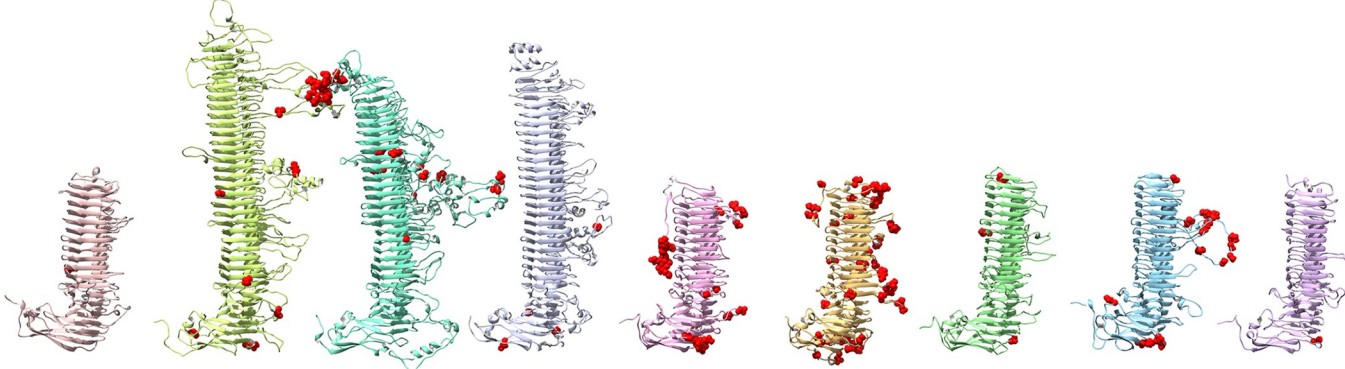

**Fig 11. Areas of high sequence variation between serovars are located primarily in the side loops.** Highlighted in red are the amino acids with less than 70% sequence conservation based on a multiple sequence alignment across the *C. trachomatis* serovars by MAFFT.

Pmp-like proteins in *Methanobacteria* (best hit, accession A0A125RDR3, E-value 1.7e-17) [45], and *Trichomonas* (best hit, accession A2E8Q2, E-value 0.004) [88]. In contrast, the *Chlamydia trachomatis* "Pmp-like proteins" Pls1, Pls2 and Pls3 reported by Jorgensen and Valdivia [86] based on marginal PSI-BLAST similarity with the PmpC passenger domain do not show any significant match with our HMM, and do not show any structural homology with existing Pmps.

Cartoon representations of autotransporters typically show the passenger domain in line with the transmembrane β-barrel sticking out from the membrane at a right angle [79, 83, 89–91]. However structural modeling of the Pmps shows a variety of potential angles between the β-barrel and the passenger domain, and molecular dynamics shows the presence of a flexible hinge between the two domains. It is not clear whether this is a general feature of autotransporters or unique to the Pmp family. The potential for passenger domains to lay flat against the outer membrane also opens the possibility for Pmp's to make longer-range protein-protein interactions with other members of the outer membrane complex and may help define their functional and cellular features in the life cycle of chlamydia.

Mutating or deleting the conserved GGA(I,L,V) and FxxN motifs has been shown to disrupt Pmp oligomerization [33] and host cell adhesion [32, 33]. Our structural models clearly show that the conserved residues of the tetrapeptide motifs are actually located on the inside of the β-helix (see Fig 5C, 5D). This suggests that the motifs are more likely to play a purely structural role in ensuring the proper folding and stacking coils of the β-helical passenger domain coils, rather than being directly involved in binding to host membranes, host proteins, or other Pmp proteins. The Pmp21-D passenger domain fragment studied in [32, 33] consist of a total of 6 β-helical coils, only the first two of which have canonical FxxN motifs, although there are stacking interactions with phenylalanines in the next two coils as well. AlphaFold2-MULTIMER predicts that the β-helices of the monomers stack N-terminal to N-terminal and C-terminal to C-terminal, forming a progressively longer beta-helical oligomer (Fig 12A). ESMFold

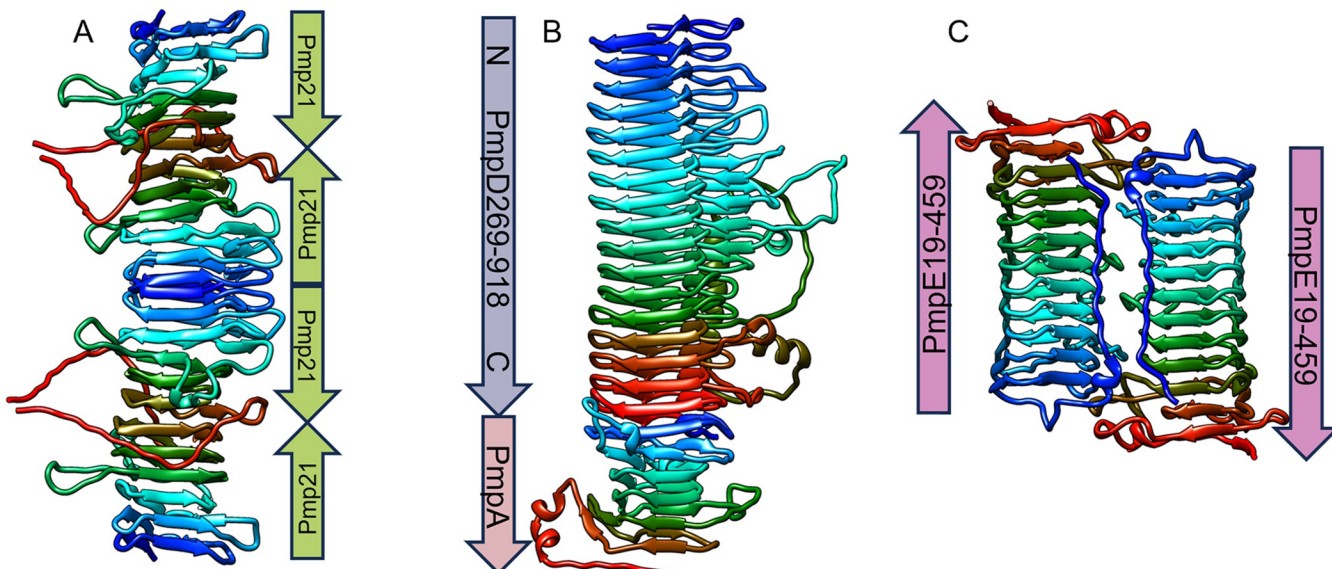

**Fig 12. Predicted Pmp oligomers.** (A) The predicted oligomer structure for Pmp21-D forms an extended β-helix, joined N-terminal to N-terminal and C-terminal to C-terminal. (B) The predicted PmpA[408-608]:PmpD[269-918] heterodimer structure for also forms an extended β-helix, with the N-terminal of PmpA[408-608] meeting up with the C-terminal of PmpD[269-918]. (C) the predicted PmpE19-459 homodimer structure shows an antiparallel configuration. All structures are shown in Rainbow coloring, with N-terminal in blue, C-terminal in red.

prediction of the Pmp21-D protein fragment with the two FxxN motifs mutated to SxxV (Pmp21-D-mut3) shows a distortion of the first two coils of the β-helix, which may inhibits stacking at the N-terminals and thus formation of longer oligomers. In addition, the first FxxN motif (FYGN) contains an outwards-facing tyrosine that is predicted by DREAMM to be a membrane-interacting residue for the wild type Pmp21-D, but not for the mutated version. So even though the phenylalanine and asparagine of the FxxN motif are internal to the β-helix, disrupting these key residues can affect both oligomerization and adhesion.

Several other Pmp passenger domain fragments have been shown to assemble into homomeric and heteromeric filaments as well [34]. We used AlphaFold2-MULTIMER to predict the dimer, trimer and tetramer structures of the PmpA, PmpD, PmpG and PmpI fragments from this study, as well as the full length PmpA-PmpI passenger domains, and the C. pneumoniae Pmp21 fragments from [32]. Preliminary results suggest that Pmp passenger domain fragments may be able to oligomerize in a number of different configurations. Fig 12B and 12C shows two of the dimer predictions with the highest model confidence: a PmpA$^{408-608}$: PmpD$^{269-918}$ heterodimer that forms a single long β-helix, and a PmpE$^{19-459}$ homodimer in an antiparallel configuration similar to that observed for the Ag43 passenger domain, involved in bacterial clumping [92].

A recent Genome-Wide Association study of *C. trachomatis* serovar G genomes identified polymorphisms in Pmp E, F, and H that are associated with rectal tissue tropism, localized specifically to the side loops in these Pmps [93]. Pmps are also known to be directly involved in invasion of host cells by binding to host EGFR in *C. pneumoniae* [94] and *C. psittaci* [95]. Although the precise mechanisms of these interactions are as yet poorly understood, the proline-rich regions and membrane interaction domains we identified in the side loops may be involved in mediating host EGFR interactions. PmpD also contains an integrin-binding RGD motif on a short loop in turn T1, that may be involved in attachment or host entry [37].

This enhanced understanding of the structural role of the tetrapeptide motifs shifts the perspective towards the areas between these conserved motifs. We have shown that the side loops jutting out from the β-helical backbone of the passenger domain contain many interesting features including protease cleavage sites, host cell adhesion, and B-cell epitopes. Conversely, T-cell epitopes are predominantly found in the β-helix itself. These results expand on the structural insights by Debrine et al [45], and highlight the modular structure encoded within Pmp passenger domains that may lend itself well to protein engineering and rational vaccine design.

The ideal *Chlamydia* vaccine would provide cross-serovar protection with strong cell-mediated and humoral responses. In this study, we extended the previously analyzed T- and B-cell epitopes in PmpD by Russi *et al.* [96] to the entire Pmp family. We are also leveraging much higher quality structure predictions that have recently become available, and especially benefit the B-cell epitope prediction. For example, several of the sequence-based B-cell epitope predictions derived by Russi et al using Bepipred [69] overlapped with the tetrapeptide motifs, which we predict are embedded in the β-helix and therefore unlikely to be accessible to antibodies. The study by Russi *et al.* focused on six peptides that included multiple B- and T-cell epitope predictions. However, our analysis shows that B- and T-cell epitopes are largely disjoint, with B-cell epitopes predominantly in side loops, while MHC-II T-cell epitopes are concentrated more inside the β-helix. For vaccine design it may therefore be more effective to choose B- and T-cell epitopes separately, rather than expect to be able to capture both with the same peptides. Given our current understanding of the structure and function of the Pmp passenger domains, a future peptide-based Pmp vaccine may need to combine parts of the β-helical scaffold that contains a good variety of T-cell epitopes, with specific side loops that carry the most promising B-cell epitopes. Conversely, it is possible that an effective Pmp vaccine will require the

recombinant expression of the whole protein. The Pmp-based vaccines tested in animal models thus far have been based almost exclusively on passenger domain fragments–often based on the known protease cleavage products, or fragments with the highest number of canonical tetrapeptide motifs. Protection elicited by the current human papilloma virus, hepatitis B virus, and SARS-CoV-2 vaccines depend on tertiary conformation of the antigen. Therefore, a full-length and native conformation of the passenger domain or the entire Pmp protein may be required for optimal protection, possibly requiring engineering out some of the cleavage sites. This paper provides an in-depth analysis of the Pmp protein family that can serve as a foundation to rational vaccine design and a structural vaccinology approach to develop an effective Pmp vaccine.

## Supporting information

**S1 File. HMM model for the longer repeat pattern we identified.**
(HMM)

**S2 File. File ZIP file containing PDB files with the RoseTTAFold structure predictions for PmpA to PmpI of *C. trachomatis* serovar E.**
(ZIP)

**S3 File. File ZIP file with multiple sequence alignment for each of the *C. trachomatis* Pmps using MAFFT.**
(ZIP)

**S1 Table. Accession numbers for the C. trachomatis, C. muridarum and C. pneumoniae Pmp proteins used in this study.**
(XLSX)

**S2 Table. ScanProsite motif hits.**
(XLSX)

**S3 Table. B-cell epitope predictions using Bepipred-2.0.**
(XLSX)

**S4 Table. B-cell epitope predictions using Discotope 2.0.**
(XLSX)

**S5 Table. MHC-II class T-cell epitope predictions using IEDB's Consensus 2.22.**
(XLSX)

**S6 Table. Membrane-penetrating residues predicted by DREAMM.**
(XLSX)

## Author Contributions

**Conceptualization:** Luis M. de la Maza, Matthew A. Coleman, Patrik D'haeseleer.

**Formal analysis:** Patrick W. Cervantes, Edmond Y. Lau, Patrik D'haeseleer.

**Funding acquisition:** Luis M. de la Maza, Matthew A. Coleman, Patrik D'haeseleer.

**Investigation:** Patrick W. Cervantes, Edmond Y. Lau, Patrik D'haeseleer.

**Methodology:** Patrik D'haeseleer.

**Project administration:** Matthew A. Coleman, Patrik D'haeseleer.

**Supervision:** Luis M. de la Maza, Matthew A. Coleman, Patrik D'haeseleer.

**Validation:** Beverly V. Robinson.

**Visualization:** Patrick W. Cervantes, Brent W. Segelke, Patrik D'haeseleer.

**Writing – original draft:** Patrick W. Cervantes, Edmond Y. Lau, Patrik D'haeseleer.

**Writing – review & editing:** Patrick W. Cervantes, Brent W. Segelke, Edmond Y. Lau, Beverly V. Robinson, Abisola Abisoye-Ogunniyan, Sukumar Pal, Luis M. de la Maza, Matthew A. Coleman, Patrik D'haeseleer.

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
