## [Decision Letter · Decision Letter 0]

11 Dec 2023

PONE-D-23-36013Sequence, Structure, and Epitope Analysis of the Polymorphic Membrane Protein Family in Chlamydia trachomatisPLOS ONE

Dear Dr. D'haeseleer,

Thank you for submitting your manuscript to Traffic. Your study has been sent to two referees for evaluation, and their reports are enclosed below. As you can see, while both referees consider the findings interesting, they also raise important points that need to be addressed before they can support publication in PLoS One. The paper by Debrine et al (doi: 10.1128/spectrum.03242-23), that was published before you submitted your MS, needs to be acknowledged. It will be useful for readers to state clearly in the discussion whether your conclusions differ or not, considering that the same prediction tools have been used. Also, as stated by reviewer 2, it is expected in the discussion (or in the results section, if helps keeping the flow) that you validate (or not) some of your structural predictions based on published functional work.

We look forward to receiving your revised manuscript.

Kind regards,

Agathe Subtil

Academic Editor

PLOS ONE

Journal Requirements:

2. Please expand the acronym  NIH/NIAID” (as indicated in your financial disclosure) so that it states the name of your funders in full.

Reviewers' comments:

Reviewer's Responses to Questions

**Comments to the Author**

1. Is the manuscript technically sound, and do the data support the conclusions?

Reviewer #1: Yes

Reviewer #2: Yes

2. Has the statistical analysis been performed appropriately and rigorously? 

Reviewer #1: Yes

Reviewer #2: I Don't Know

3. Have the authors made all data underlying the findings in their manuscript fully available?

Reviewer #1: Yes

Reviewer #2: Yes

4. Is the manuscript presented in an intelligible fashion and written in standard English?

Reviewer #1: Yes

Reviewer #2: Yes

5. Review Comments to the Author

Reviewer #1: The manuscript by Cervantes et al provides an excellent description of in-silico-based analysis of the chlamydial Pmp family of proteins. The depth of analysis is considerable and the paper is well written. I have no significant criticisms of the work. The molecular dynamics, epitope prediction, and hinge analysis are particularly unique.

A similar paper has recently been published that describes an Alphafold-based analysis of this family of proteins. This manuscript and that paper look at the same issue from slightly different perspectives and both will be useful to the research community. It is suggested that these authors cite the other manuscript and perhaps briefly describe differences between the documents, with a goal of helping researchers get a wider perspective on predicted Pmp structure. There are data in that manuscript that differ slightly from this one, and certain subjects are addressed by one and not the other one. Both documents will be valuable to people who wish to explore chlamydial Pmp structure.

Minor comments about the writing:

Lines 183 vs. 185, and elsewhere… the designation of I,L,V or ILV needs to be consistent.

Perhaps the MOMP presentation in the Introduction is too drawn out. Lines 32-43 could probably be condensed to one or two sentences.

Lines 80-83: A distinction between the sequence similarity in the beta barrel vs the beta helical domain, across the Pmp proteins, might be useful here.

Reviewer #2: Review of MS PONE-D-23-36013 by Cervantes et al.

This MS utilizes protein structure prediction algorithms to visualize, for the first time, the Pmp family from C. trachomatis Serovar E. A special focus has been put onto the tetrapeptide motifs, GGA(I,L,V) and FxxN.

The MS is interesting and adds to our understanding of Pmp protein function. My main critics is that the authors do not try to compare their Pmp structures with published functional data on Pmp proteins and protein fragments. We have published for C. pneumoniae Pmp21, that different protein fragments have different adhesion capacities and that protein fragments with 2 motifs, when point-mutated, lose this capacity (Mölleken et al ). Our study on C. trachomatis Pmps shows that certain motif-rich and motif-poor Pmp protein fragments have a different capacity for adhesion and oligomerization (Favaroni 2021). Finally, we have used large segments of certain C. trachomatis Pmp proteins for a vaccine study (Lanfermann et al; doi.org/10.3390/vaccines9060609). Finally, our biochemical work on a Pmp21 protein fragment has revealed that this protein domain forms oligomers that adopt an amyloid-like structure comparable in size and shape to protofibrils of the amyloid protein Abeta42 (Luczak et al). It would be really interesting to compare the detailed structures presented in this MS with some of the functional data published.

The introduction is detailed but needs a few corrections:

Line 47: “….viable vaccine candidates…”. The authors missed one recent publication which convincingly shows that a mix of different recombinant C. trachomatis Pmps induces a high degree of cross-serovar protection against urogenital and ocular strains of C. trachomatis: Lanfermann et al., DOI: 10.3390/vaccines9060609.

Line 56: “Pmps are autotransporters”. That is not true. Pmps are predicted to be autotransporters. So far no experimental prove. Needs to be corrected.

Fig. 6 shows a molecular dynamics simulation of PmpE with the β-barrel domain embedded in a POPC bilayer. I am wondering whether the simulation gives a similar result, when the authors take into account that a typical outer membrane of a G- - negative bacterium is very asymmetrical, with phospholipids in the inner leaflet and LPS in the outer leaflet, plus all of the transmembrane proteins with their β-barrel fold (outer membrane proteins (OMPs)) and the lipoproteins.

Fig. 7. I could not find a figure legend.

Line 359. The authors define proline-rich regions (PRRs) as having at least a P-X0-2-P motive (see Fig. 9 legend). What is the basis of this definition?

Fig. 9. The legend says: “…A few proline-rich regions in side loops of PmpC, PmpG, PmpH and PmpI are labeled.” . To make this point stronger, the authors should label all proline-rich regions - not only a few.

Johannes Hegemann

6. PLOS authors have the option to publish the peer review history of their article (what does this mean?). If published, this will include your full peer review and any attached files.

Reviewer #1: No

Reviewer #2: **Yes: **Johannes H. Hegemann

---

## [Author Response · Author response to Decision Letter 0]

6 Mar 2024

Reviewer #1: The manuscript by Cervantes et al provides an excellent description of in-silico-based analysis of the chlamydial Pmp family of proteins. The depth of analysis is considerable and the paper is well written. I have no significant criticisms of the work. The molecular dynamics, epitope prediction, and hinge analysis are particularly unique.

A similar paper has recently been published that describes an Alphafold-based analysis of this family of proteins. This manuscript and that paper look at the same issue from slightly different perspectives and both will be useful to the research community. It is suggested that these authors cite the other manuscript and perhaps briefly describe differences between the documents, with a goal of helping researchers get a wider perspective on predicted Pmp structure. There are data in that manuscript that differ slightly from this one, and certain subjects are addressed by one and not the other one. Both documents will be valuable to people who wish to explore chlamydial Pmp structure.

We thank the reviewer for pointing out this paper. We had actually shared a copy of the preprint of our manuscript with the authors earlier, but had missed the publication of their paper. We have added a number of references throughout the manuscript. 

Minor comments about the writing:

Lines 183 vs. 185, and elsewhere… the designation of I,L,V or ILV needs to be consistent.

After reviewing dozens of papers on the Pmp repeat motifs, we found that “GGA(I,L,V)” is the most common notation used in text, and we have tried to use it consistently throughout the paper. We did correct one instance on line 74. On line 185, we list the corresponding pattern in PROSITE notation to be used on the ScanProsite search engine, which does look significantly different. We have edited that line to better explain the difference in notation.

Perhaps the MOMP presentation in the Introduction is too drawn out. Lines 32-43 could probably be condensed to one or two sentences.

We have left out the mention of immunotyping using MOMP, which is not really relevant here. 

Lines 80-83: A distinction between the sequence similarity in the beta barrel vs the beta helical domain, across the Pmp proteins, might be useful here.

We’ve updated the text to clarify that the majority of Pmps (except PmpB vs PmpC) share less than 30% amino acid identity, and that sequence identity is similar in the beta barrel versus the passenger domain. We tend to think of the beta barrel as well conserved, but that is mainly because there is a reliable Pfam domain for it. 

Reviewer #2: Review of MS PONE-D-23-36013 by Cervantes et al.

This MS utilizes protein structure prediction algorithms to visualize, for the first time, the Pmp family from C. trachomatis Serovar E. A special focus has been put onto the tetrapeptide motifs, GGA(I,L,V) and FxxN.

The MS is interesting and adds to our understanding of Pmp protein function. My main critics is that the authors do not try to compare their Pmp structures with published functional data on Pmp proteins and protein fragments. We have published for C. pneumoniae Pmp21, that different protein fragments have different adhesion capacities and that protein fragments with 2 motifs, when point-mutated, lose this capacity (Mölleken et al ). Our study on C. trachomatis Pmps shows that certain motif-rich and motif-poor Pmp protein fragments have a different capacity for adhesion and oligomerization (Favaroni 2021). Finally, we have used large segments of certain C. trachomatis Pmp proteins for a vaccine study (Lanfermann et al; doi.org/10.3390/vaccines9060609). Finally, our biochemical work on a Pmp21 protein fragment has revealed that this protein domain forms oligomers that adopt an amyloid-like structure comparable in size and shape to protofibrils of the amyloid protein Abeta42 (Luczak et al). It would be really interesting to compare the detailed structures presented in this MS with some of the functional data published.

We have included a section on Pmp21-D, using ESMfold to predict the effect of the FxxN mutations on the structure, AlphaFold-MULTIMER to predict how this may affect oligomerization, and DREAMM to show the effect on a predicted membrane-interacting tyrosine within the first mutated FxxN. 

We had hoped to find a simple oligomerization mechanism across all Pmp passenger domain fragments, like the one observed for the H.influenza Hap adhesin (see https://www.ncbi.nlm.nih.gov/pmc/articles/PMC3173798/figure/f5/). AlphaFold-MULTIMER predictions of oligomer structure of the PmpA-PmpI passenger domains, as well as the Pmp21, PmpA, PmpD, PmpG and PmpI fragments described in these papers do not tend to yield a consistent oligomerization pattern. Out of the 34 AlphaFold-MULTIMER dimer, trimer and tetramer runs, only 10 yielded moderate to good confidence structures (pTM = 0.685 – 0.874), including 7 where the two beta-helices were in line with each other (4 C-terminal to C-terminal, 2 N-terminal to N-terminal, and 1 N-terminal to C-terminal), and 3 with the beta-helices next to each other (1 parallel, 2 antiparallel). Interestingly, the PmpA+G, PmpA+I, and especially PmpA+D heterodimers from Favaroni 2021 all show higher model confidence scores than their respective homodimers, in line with the oligomerization results from that paper. Also note that the two “motif-poor” Pmp protein fragments in Favaroni 2021 (A and G) are actually predicted to have an equivalent number of beta-helical coils as the “motif-poor” ones, with an average of around 38 amino acids per coil. The main difference lies in the fact that the “motif-poor” fragments are shorter, and overlap the Middle domain, which has a different sequence specificity (but still largely beta-helical structure). We have included a figure with the Pmp21-D predicted tetramer model, and the two most high-confidence dimer models (the PmpA+D heterodimer, and PmpE homodimer).

The introduction is detailed but needs a few corrections:

Line 47: “….viable vaccine candidates…”. The authors missed one recent publication which convincingly shows that a mix of different recombinant C. trachomatis Pmps induces a high degree of cross-serovar protection against urogenital and ocular strains of C. trachomatis: Lanfermann et al., DOI: 10.3390/vaccines9060609.

 Reference has been added 

Line 56: “Pmps are autotransporters”. That is not true. Pmps are predicted to be autotransporters. So far no experimental prove. Needs to be corrected.

Corrected.

Fig. 6 shows a molecular dynamics simulation of PmpE with the β-barrel domain embedded in a POPC bilayer. I am wondering whether the simulation gives a similar result, when the authors take into account that a typical outer membrane of a G- - negative bacterium is very asymmetrical, with phospholipids in the inner leaflet and LPS in the outer leaflet, plus all of the transmembrane proteins with their β-barrel fold (outer membrane proteins (OMPs)) and the lipoproteins.

The main purpose of doing the molecular dynamics simulations was to explain why many of the structure predictions show the passenger domain at a physiologically unrealistic sharp angle to the beta barrel, where it would intersect the membrane if the beta barrel was embedded in the OM. Clearly the presence of LPS and abundant transmembrane proteins such as MOMP would affect the angle as well, but such simulations would be much more expensive and may not add much additional insight to the topic in this manuscript. We hope to include some of this in a future paper on the Chlamydia Outer Membrane Complex. 

Fig. 7. I could not find a figure legend.

We have reinserted the missing figure legend.

Line 359. The authors define proline-rich regions (PRRs) as having at least a P-X0-2-P motive (see Fig. 9 legend). What is the basis of this definition?

Fig. 9. The legend says: “…A few proline-rich regions in side loops of PmpC, PmpG, PmpH and PmpI are labeled.” . To make this point stronger, the authors should label all proline-rich regions - not only a few.

We were not able to find a unique definition of what constitutes the minimum requirement for a proline-rich region. Examining the wide range of different repeating and non-repeating proline-rich regions listed in Williamson 1994, the vast majority seem to contain at least four prolines, separated by no more than four other amino acids. We’ve decided to adopt this as our criterion for Figure 9, which also allows us to label all the corresponding regions on that figure.

---

## [Decision Letter · Decision Letter 1]

16 Apr 2024

PONE-D-23-36013R1Sequence, Structure, and Epitope Analysis of the Polymorphic Membrane Protein Family in Chlamydia trachomatisPLOS ONE

Dear Dr. D'haeseleer,

Thank you for submitting your revised manuscript to PLOS ONE. You have satisfactorily responded to all the reviewers's comments. However, reviewer 2 made a finale point that I would appreciate you to address by sending a revised version (very minor revision).  

Please submit your revision by May 31 2024 11:59PM.

We look forward to receiving your revised manuscript.

Kind regards,

Agathe Subtil

Academic Editor

PLOS ONE

Journal Requirements:

Additional Editor Comments:

Congratulation for this paper. PLease take into account the comments from reviewer 2 for minor revision of your manuscript.

Reviewers' comments:

Reviewer's Responses to Questions

**Comments to the Author**

1. If the authors have adequately addressed your comments raised in a previous round of review and you feel that this manuscript is now acceptable for publication, you may indicate that here to bypass the “Comments to the Author” section, enter your conflict of interest statement in the “Confidential to Editor” section, and submit your "Accept" recommendation.

Reviewer #2: All comments have been addressed

2. Is the manuscript technically sound, and do the data support the conclusions?

Reviewer #2: Yes

3. Has the statistical analysis been performed appropriately and rigorously? 

Reviewer #2: I Don't Know

4. Have the authors made all data underlying the findings in their manuscript fully available?

Reviewer #2: Yes

5. Is the manuscript presented in an intelligible fashion and written in standard English?

Reviewer #2: Yes

6. Review Comments to the Author

Reviewer #2: Review of MS PONE-D-23-36013_R1 by Cervantes et al.

In the revised MS, the authors dealt satisfactorily with all the points I had raised. In particular, I like the data and the discussion about the oligomerization potential of Pmp proteins.

One important aspect I did not mention in my first review concerns the terminology: The authors use a structure prediction program as a basis for their work. Therefore, all conclusions from these structural predictions are also predictions and not experimentally documented facts. This should be clearly indicated throughout the MS, e.g.

- the title of this MS omits this fact,

- Line 11. Under ´Keywords` no mentioning of the use of a structure prediction program….

- in the second line of the abstract (lines 15-16) the authors state “Here we show that the 16 characteristic GGA(I,L,V) and FxxN tetrapeptide repeats fit into a larger repeat sequence….” . Up to here, no mentioning of a structure prediction program. I suggest starting the abstract with sentence # three of the current abstract (Analysis of the protein using structure prediction algorithms provided novel insight to the chlamydial Pmp family of proteins.).

- line 23. The authors state `This structural analysis….` . It is a predicted structure…

Minor points:

- Line 54. The Greek symbol for beta is missing.

- Figures 1+2. Some of the text in the figures are not readable…..

Johannes Hegemann

7. PLOS authors have the option to publish the peer review history of their article (what does this mean?). If published, this will include your full peer review and any attached files.

Reviewer #2: **Yes: **Johannes Hegemann

---

## [Author Response · Author response to Decision Letter 1]

11 May 2024

Reviewer #2:

In the revised MS, the authors dealt satisfactorily with all the points I had raised. In particular, I like the data and the discussion about the oligomerization potential of Pmp proteins.

One important aspect I did not mention in my first review concerns the terminology: The authors use a structure prediction program as a basis for their work. Therefore, all conclusions from these structural predictions are also predictions and not experimentally documented facts. This should be clearly indicated throughout the MS, e.g.

- the title of this MS omits this fact,

- Line 11. Under ´Keywords` no mentioning of the use of a structure prediction program….

Note that structure prediction was explicitly mentioned in the 2nd and 3rd sentences of the abstract. But for completeness, we have included it in the title and keywords, as suggested.

- in the second line of the abstract (lines 15-16) the authors state “Here we show that the 16 characteristic GGA(I,L,V) and FxxN tetrapeptide repeats fit into a larger repeat sequence….” . Up to here, no mentioning of a structure prediction program. I suggest starting the abstract with sentence # three of the current abstract (Analysis of the protein using structure prediction algorithms provided novel insight to the chlamydial Pmp family of proteins.).

The analysis of the tetrapepdtide motifs (section 4.1), showing that they fit into a longer repeat pattern is indeed purely sequence based. It is rather surprising that this was not caught by Grimwood and Stephens back in 1999 when they claimed that the motifs did “not appear to follow any characteristic spacing pattern”, because to our eyes the alternating pattern and semi-regular spacing on Pmp sequence maps was quite striking. We feel that the flow of the manuscript works better by starting with the purely sequence based analysis, then follow it up with how the longer sequence repeat pattern that we identified matches with the rungs of the beta-helix in the predicted structures. We decided to keep the order in the abstract the same as the order in which these analyses are presented in the Results section.

- line 23. The authors state `This structural analysis….` . It is a predicted structure…

We have omitted “structural” here, because the analysis is based both on sequence and predicted structure

Minor points:

- Line 54. The Greek symbol for beta is missing.

We could not find a missing beta symbol on line 54, or anywhere else in this paragraph.

- Figures 1+2. Some of the text in the figures are not readable…..

We recreated Fig 1 by hand in vector graphics with slightly larger font sizes, so the text should be both larger and sharper. Likewise, we recreated the table with HMM occupancy / insertion frequencies and insertion lengths in the Fig 2C and 2D sequence logos. The reader may still need to download the full size image if they really want to read the HMM weights, but at least the numbers are larger and far more crisp in the new version.

---

## [Editor Report · Decision Letter 2]

14 May 2024

Sequence, Structure Prediction, and Epitope Analysis of the Polymorphic Membrane Protein Family in Chlamydia trachomatis

PONE-D-23-36013R2

Dear Dr. D'haeseleer,

We’re pleased to inform you that your manuscript has been judged scientifically suitable for publication and will be formally accepted for publication once it meets all outstanding technical requirements.

Kind regards,

Agathe Subtil

Academic Editor

PLOS ONE
---

## [Editor Report · Acceptance letter]

31 May 2024

PONE-D-23-36013R2 

PLOS ONE

Dear Dr. D'haeseleer, 

I'm pleased to inform you that your manuscript has been deemed suitable for publication in PLOS ONE. Congratulations! Your manuscript is now being handed over to our production team.

Kind regards, 

on behalf of

Dr. Agathe Subtil 

Academic Editor

PLOS ONE